# Characterization of High-Risk HPV/EBV Co-Presence in Pre-Malignant Cervical Lesions and Squamous Cell Carcinomas

**DOI:** 10.3390/microorganisms10050888

**Published:** 2022-04-24

**Authors:** Rancés Blanco, Diego Carrillo-Beltrán, Juan P. Muñoz, Julio C. Osorio, Julio C. Tapia, Verónica A. Burzio, Iván Gallegos, Gloria M. Calaf, Paola Chabay, Francisco Aguayo

**Affiliations:** 1Laboratorio de Oncovirología, Programa de Virología, Instituto de Ciencias Biomédicas (ICBM), Facultad de Medicina, Universidad de Chile, Santiago 8380000, Chile; rancesblanco1976@gmail.com (R.B.); diegocb17@hotmail.com (D.C.-B.); 2Instituto de Alta Investigación, Universidad de Tarapaca, Arica 1000000, Chile; juanpablomunozbarrera@gmail.com (J.P.M.); gmc@cumc.columbia.edu (G.M.C.); 3Population Registry of Cali, Department of Pathology, Universidad del Valle, Cali 760042, Colombia; cejulio704@gmail.com; 4Laboratorio de Transformación Celular, Programa Biología Celular y Molecular, Instituto de Ciencias Biomédicas (ICBM), Facultad de Medicina, Universidad de Chile, Santiago 8380000, Chile; jtapiapineda@uchile.cl; 5Facultad de Ciencias de la Vida, Universidad Andrés Bello, Fundación Ciencia & Vida, Andes Biotechnologies SpA, Santiago 7780272, Chile; vburzio@unab.cl; 6Servicio de Anatomía Patológica, Hospital Clínico de la Universidad de Chile, Santiago 8380000, Chile; igallegos@hcuch.cl; 7Laboratory of Molecular Biology, Pathology Division, Multidisciplinary Institute for Investigation in Pediatric Pathologies (IMIPP-CONICET-GCBA), Ricardo Gutiérrez Children’s Hospital, Buenos Aires C1425EFD, Argentina; paola_chabay@yahoo.com.ar; 8Advanced Center for Chronic Diseases (ACCDiS), Universidad de Chile, Santiago 8330024, Chile

**Keywords:** Epstein–Barr virus, human papillomavirus, BamHI-A rightward frame 1 expression, cervical cancer

## Abstract

High-risk human papillomaviruses (HR-HPVs) are the etiological agents of cervical cancer. However, a low proportion of HR-HPV-infected women finally develop this cancer, which suggests the involvement of additional cofactors. Epstein–Barr virus (EBV) has been detected in cervical squamous cell carcinomas (SCCs) as well as in low- (LSIL) and high-grade (HSIL) squamous intraepithelial lesions, although its role is unknown. In this study, we characterized HR-HPV/EBV co-presence and viral gene expression in LSIL (*n* = 22), HSIL (*n* = 52), and SCC (*n* = 19) from Chilean women. Additionally, phenotypic changes were evaluated in cervical cancer cells ectopically expressing BamHI-A Rightward Frame 1 (BARF1). BARF1 is a lytic gene also expressed in EBV-positive epithelial tumors during the EBV latency program. HPV was detected in 6/22 (27.3%) LSIL, 38/52 (73.1%) HSIL, and 15/19 (78.9%) SCC cases (*p* < 0.001). On the other hand, EBV was detected in 16/22 (72.7%) LSIL, 27/52 (51.9%) HSIL, and 13/19 (68.4%) SCC cases (*p* = 0.177). HR-HPV/EBV co-presence was detected in 3/22 (13.6%) LSIL, 17/52 (32.7%) HSIL, and 11/19 (57.9%) SCC cases (*p* = 0.020). Additionally, BARF1 transcripts were detected in 37/55 (67.3%) of EBV positive cases and in 19/30 (63.3%) of HR-HPV/EBV positive cases. Increased proliferation, migration, and epithelial-mesenchymal transition (EMT) was observed in cervical cancer cells expressing BARF1. Thus, both EBV and BARF1 transcripts are detected in low- and high-grade cervical lesions as well as in cervical carcinomas. In addition, BARF1 can modulate the tumor behavior in cervical cancer cells, suggesting a role in increasing tumor aggressiveness.

## 1. Introduction

Cervical cancer is the fourth most commonly diagnosed malignant tumor in women worldwide and also the fourth cause of death [1]. In 2018, an estimated 569,847 new cases of cervical cancer and 311,365 deaths attributable to this malignancy were reported, with a marked prevalence in low- and middle-income countries [2]. In Chile, cervical cancer is the sixth most common diagnosed tumor among women and the second most frequent type of cancer in women between the ages of 20 and 44 years. It caused 725 deaths among females in 2018, mainly belonging to low socio-economic levels. Most cervical tumors (≥95%) arise from epithelial cells, and 80–90% of them are squamous cell carcinomas (SCCs) [3,4].

Persistent human papillomavirus (HPV) infection is the most important risk factor for development of cervical [5] and penile [6] SCCs, as well as a subset of vulvar carcinomas [7]. A group of HPV types (16, 18, 31, 33, 34, 35, 39, 45, 51, 52, 56, 58, 59, 66, 68, and 70), so-called high-risk (HR)-HPVs, are involved in the development of 99.7% of cervical carcinomas worldwide. HPV16 and HPV18 play an important role in the development of squamous intraepithelial lesions (SILs) [8]. Furthermore, approximately 70% of cervical SCCs are caused solely by HPV16 or HPV18 types [9,10]. The overexpression of HR-HPV E6 and E7 oncoproteins in the cervical epithelium is an essential event for carcinogenesis [11,12]. These oncoproteins target p53 and pRB tumor suppressor proteins, evading apoptosis and disrupting the cell cycle in infected epithelial cells, respectively [13,14]. However, HR-HPV infection is insufficient for the development of cervical cancer. In fact, it has been reported that 50% of low-grade squamous intraepithelial lesions (LSIL) associated with HR-HPV infection (e.g., HPV16 or HPV18) regress to normal cytology [15], while only 3.6% of these precancerous cases progress to high-grade squamous intraepithelial lesions (HSIL) and potentially cause cancer [16]. Accordingly, other host and/or environmental factors are required for promoting the progression of cervical lesions. In this regard, the potential contribution of additional viral infections to cervical cancer progression is still unclear [17,18].

Epstein–Barr virus (EBV), formerly named human gammaherpesvirus-4 (HHV-4), is a member of the *Herpesviridae* family (reviewed in [19]) that infects more than 90% of the human population worldwide [20,21]. This virus is etiologically associated with a variety of tumors of both lymphoid and epithelial origin, including some lymphomas, nasopharyngeal carcinoma (NPC) and a subset of gastric cancers (GC) [22,23,24,25]. Additionally, EBV infection has been suggested to be a potential cofactor for the development and progression of cervical carcinoma. For instance, HR-HPV/EBV coinfection was shown to be increased in HSIL and cervical carcinomas when compared with LSIL and non-premalignant lesions [26]. Moreover, in EBV-positive cervical lesions it was found an increase in the methylation status of the E-cadherin (CDH1) gene promoter region, which was related with the epithelial-to-mesenchymal transition (EMT) [27]. Even though EBV latency is a requisite for EBV-driven tumors, which involves restricted and regulated viral gene expression, in recent years the contribution of EBV lytic genes to malignant transformation has been proposed [28]. Among lytic genes, the BamHI-A rightward frame 1 (BARF1) (reviewed in [29]) is mainly expressed in epithelial malignancies, although its presence has also been reported in lymphomas [30,31]. In fact, increased BARF1 expression has been detected in NPC and EBV-associated GC (EBVaGC) during the EBV latency program, which has been characterized as an exclusively epithelial oncoprotein [30,31]. However, to the best of our knowledge, only one previous study reported the expression of BARF1 in cervical carcinomas [32].

In the present study, we evaluated the prevalence of HR-HPV/EBV co-presence in premalignant lesions and cervical carcinomas from Chilean patients. Additionally, the contribution of EBV BARF1 to the aggressiveness of HPV-positive cervical cancer cells was assessed. The capacity of EBV BARF1 to induce EMT in cervical cancer cells was also evaluated by means of E-cadherin and ZEB1 protein levels.

## 2. Materials and Methods

### 2.1. Tissue Samples

Seventy-seven premalignant and 25 malignant cervical lesions from Chilean women collected between 2005 and 2010 were obtained from the Pathology Department at the Hospital Clínico “Dr. José Joaquín Aguirre” from the University of Chile. All samples were formalin-fixed and paraffin-embedded (FFPE) tissues, which were processed following the standard histological procedures. Clinical records were revised and the totality of samples with diagnosis of SIL or SCC of the uterine cervix was primarily selected. The histological diagnosis was confirmed by two experienced pathologists (C.V. and I.G.), based on hematoxylin and eosin staining. Cases that did not match the initial diagnosis or without paraffin blocks available for additional procedures were excluded from the study. This study was approved by the Ethical Committee Board of both the Hospital Clínico “Dr. José Joaquín Aguirre” and the Faculty of Medicine, Universidad de Chile (approval code No. 061-2019).

### 2.2. DNA Extraction

DNA from tissue samples was extracted as previously described [33]. Briefly, 10-µm sections from each FFPE sample were treated with digestion buffer containing 10 mM Tris–HCl pH 7.4, 0.5 mg/mL proteinase K, and 0.4% Tween 20. The specimens were incubated at 56 °C overnight under constant shaking, followed by incubation at 95 °C for 10 min. The samples were then immediately centrifuged at 14,000 rpm for 2 min and maintained on ice. The aqueous phase was transferred to a new tube and stored at −20 °C until use.

### 2.3. Polymerase Chain Reaction (PCR)

The PCR reaction mixture was prepared in a total volume of 25 µL, which contained 12.5 µL of 2X GoTaq^®^ G2 Green Master Mix (Promega, Madison, WI, USA), 0.5–0.625 μL of 20 μM forward and reverse primers, 5 μL of template DNA, and an adequate volume of nuclease-free water. Primer sequences and PCR conditions for each gene fragment are shown in Appendix A. For DNA quality determination, a β-globin gene fragment was amplified. The amplification products were stained with SafeView Plus^TM^ (ABM, Vancouver, BC, Canada), analyzed by 2.5% agarose gel electrophoresis and visualized by UV transillumination (Vilber Lourmat). The Accuruler 100 bp plus DNA ladder (MaestroGen Inc., Hsinchu, Taiwan) was used as DNA molecular weight standard control. DNA extracted from HPV16-positive CaSki cells (ATCC^®^ CRL-1550^TM^) or from a clinical sample of known positivity for EBV were used as positive controls. Digestion buffer and nuclease-free water were used as negative controls.

### 2.4. HPV Genotyping

Amplified PCR products obtained from HPV-positive specimens were sent to Macrogen Co. Ltd. (Seoul, Korea) for viral DNA sequencing. When necessary, PCR products were purified using the Wizard SV Gel and PCR clean-up kit (Promega, Madison, WI, USA) to avoid potential sample contamination. For genotyping, the sequences obtained with GP5+ and GP6+ primers were manually depurated from raw chromatograms and compared to known HPV DNA sequences stored in the nucleotide collection (nr/nt) database using the BlastN algorithm (https://blast.ncbi.nlm.nih.gov/Blast.cgi?PAGE_TYPE=BlastSearch) (Accessed date: 19 February 2021).

### 2.5. In Situ Hybridization for EBV

Epstein–Barr encoded early RNAs (EBERs) were detected by in situ hybridization with the ZytoFast EBV Probe (Digoxigenin-labeled) reagent and the ZytoFast PLUS CISH Implementation Kit (HRP-DAB) (ZytoVision, Bremerhaven, Germany) following the manufacturer’s protocols with minor modifications as follows: (1) denaturation was done using an oven instead of on a hot plate or hybridizer and (2) 8 µL probe solution was applied to each specimen instead of 10 µL. For verification of cellular mRNA integrity, the ZytoFast 28S rRNA (+) control probe was used. A known EBV-positive Hodgkin lymphoma case and Raji cells (derived from a patient with Burkitt’s lymphoma) were used as positive controls. A positive result was considered when a brown color localized in premalignant or malignant epithelial cells was observed.

### 2.6. Reverse Transcriptase-PCR

Total RNA from tissue samples was isolated using the High Pure FFPET RNA Isolation Kit (Roche Molecular Systems, Inc., Pleasanton, CA, USA) according to manufacturer’s instructions. For cDNA synthesis, a reaction mix containing 2 µg of pure total RNA, 1 U of RNase inhibitor (Promega, Madison, WI, USA), 0.04 µg/µL of random primers (Promega, Madison, WI, USA), 2 mM of dNTP mix (Promega, Madison, WI, USA) and 10 U of Moloney Murine Leukemia Virus reverse transcriptase (M-MLV RT) (Promega, Madison, WI, USA), in a total reaction volume of 20 µL, was incubated for 1 h at 37 °C and stored at −20 °C until use. For cDNA amplification, 25 µL-reaction mixtures containing 12.5 µL of 2X GoTaq^®^ G2 Green Master Mix (Promega, Madison, WI, USA), 0.5 µL of 20 µM forward and reverse primers (specific for BARF1 and HPV16 E6 transcripts), 10.5 µL of RNase-free water (Promega Corporation, Madison, WI, USA), and 1 µL of cDNA were prepared. Endogenous β-actin mRNA levels were used for normalization of RNA expression.

### 2.7. Cell Culture and Transfection

SiHa (HTB-35^TM^) and CaSki (CRL-1550^TM^) cervical carcinoma cells were obtained from the American Type Culture Collection (ATCC; Manassas, VA, USA) and cultured in RPMI-1640 basal medium (Gibco, Carlsbad, CA, USA) supplemented with 10% heat-inactivated fetal bovine serum (FBS) (Hyclone, Fremont, CA, USA), 100 U/mL penicillin, 100 g/mL streptomycin, and 0.25 μg/mL amphotericin B (Gibco, Carlsbad, CA, USA). Cells were maintained at 37 °C in a humidified atmosphere containing 5% CO_2_. For passaging, cell monolayers were washed with sterile 1X phosphate-buffered saline pH 7.4 and detached in 1X trypsin-EDTA (Gibco, Carlsbad, CA, USA) for 5 min. Cells were periodically tested for mycoplasma contamination by PCR. SiHa and CaSki cells were seeded into 6-well plates at a density of 0.5 × 10^6^ cells per well and on the next day transfected with 30 µM MSCV (Addgene, Plasmid #41033, depositing lab: Wade Harpe) or MSCVBARF1 (Addgene, Plasmid #37922, depositing lab: Karl Munger) plasmids using Lipofectamine^®^ 2000 (Invitrogen, Carlsbad, CA, USA) according to manufacturer’s protocol. Cells were maintained in culture medium without antibiotics for 12–18 h, after which transfected cells were selected by addition of 0.2 µg/mL puromycin for seven days (Gibco, Carlsbad, CA, USA).

### 2.8. Phosphoproteomic NF-kB Array

Empty vector and BARF1-transfected SiHa cells were grown to 90% confluence in 10 cm plates and subjected to serum starvation for 24 h. Cells were then collected by centrifugation and washed once in phosphate buffered saline (PBS). Cell pellets were suspended in extraction lysis buffer and incubated for 20 min at 4 °C. Protein concentration was determined using the Pierce^TM^ BCA protein assay kit (Thermo Scientific, Rockford, IL, USA) according to manufacturer’s instructions. Screening of different proteins in cell lysates was performed with a Proteome profiler array kit (ARY029, R&D Systems), for the parallel determination of relative levels of phosphorylation of NF-kB. The array allows determination of the relative phosphorylation of P53^S46^, RelA/p65^S529^, STAT1^Y701^, and STAT2^Y689^. Horseradish peroxidase substrate (Millipore Corporation, Burlington, VT, USA) was used to detect protein signal and data was captured by exposure to Fujifilm Light films. Films were analyzed with the NIH ImageJ software.

### 2.9. Cell Migration Assay

Cell migration capacity was assessed by Boyden chamber assay (Transwell migration assay) as previously reported by our group [34]. Briefly, the bottom side of transwell upper chambers (Corning, New York, NY, USA) was coated with 2 µg/mL fibronectin (Thermo Fisher Scientific, Inc., MS, USA) and incubated overnight at 4 °C. Then, 7000 SiHa and CaSki cells were seeded inside the inserts in 200 µL of serum-free RPMI-1640 media, and 500 µL of RPMI-1640 supplemented with 10% FBS was added to each plate well. Cells were allowed to migrate for 7 h at 37 °C and 5% CO_2_. Then, migrated cells were fixed and stained in a solution of 0.5% crystal violet in 20% methanol for 1 h. Unmigrated cells were scraped from the upper chambers using cotton swabs. Migrated cells were counted in seven high-power fields (400×) for three independent experiments.

### 2.10. Cell Proliferation Assay

For evaluation of cell proliferation we used the MTS (3-(4,5-dimethylthiazol-2-yl)-5-(3-carboxymethoxyphenyl)-2-(4-sulfophenyl)-2H-tetrazolium) (Promega Corporation, Madison, WI, USA) colorimetric assay [34]. Briefly, 3 × 10^4^ cells were cultured in 96-well flat-bottom cell culture microplates in RPMI-1640 supplemented with 10% FBS. At 24, 48, 72, and 96 h, 30 µL of MTS reagent was added to each well and plates were incubated for 3 h at 37 °C and 5% CO_2_. Finally, absorbance was measured at 492 nm using a Synergy 2 spectrophotometer (BioTek Instruments Inc., Winooski, VT, USA).

### 2.11. Anchorage-Independent Growth Assay

To evaluate cell colony formation in soft agar, the procedure previously reported by our group was used [35]. SiHa and CaSki cells (5 × 10^3^) were suspended in 0.33% Bacto-agar (BD Biosciences, Heidelberg, Germany) dissolved in RPMI-1640 supplemented with 12.5% FBS. Cell suspensions were added to 6-well plates containing 2 mL of 0.5% agar dissolved in the same medium and cultured at 37 °C under normal conditions. Fresh RPMI-1640/10% FBS (0.5 mL) was added to each well twice a week. After 3 weeks, cells were stained in 0.005% crystal violet dissolved in 20% methanol for 1 h at room temperature. Finally, cell colonies were photographed with a Nikon D5100 camera and differences between empty vector and BARF1-transfected cells were visually estimated.

### 2.12. Western Blot

Protein lysates obtained from empty vector and BARF1-transfected cells were extracted with RIPA 1X lysis buffer (Abcam, Cambridge, UK) containing protease and phosphatase inhibitor cocktail (Roche, Basel, Switzerland). Suspensions were incubated at 4 °C for 10 min, sonicated in an ice bath for 20 s, and then centrifuged at 14,000× *g* for 15 min. Protein concentration was quantified with the Pierce^TM^ BCA protein assay kit (Thermo Scientific, Rockford, IL, USA). Thirty micrograms of total protein were loaded per well and separated by SDS-PAGE on 12% gels. Proteins were then transferred by electroblotting to Hybond-P ECL membranes (Amersham, Piscataway, NJ, USA) using a pH 8.3 Tris-glycine transfer buffer (20 mM Tris, 150 mM glycine, 20% methanol) and a Trans–Blot^®^ SD semi–dry electrophoretic transfer cell (Bio-Rad, Hercules, CA, USA). Membranes were blocked in 5% bovine serum albumin/0.5% Tween-20 in Tris buffered saline pH 7.6 (TBS) for 1 h at room temperature and then incubated overnight at 4 °C with mouse anti-human E-cadherin monoclonal antibody (Santa Cruz Biotechnology, Inc., Dallas, TX, USA) or rabbit anti-human ZEB1 polyclonal antibody (Thermo Fisher Scientific, Inc., MS, USA) diluted 1:1000 in TBS/Tween 20 (TBS–T20). After three washes in TBS–T20, membranes were incubated either with anti-mouse IgG (BD Pharmingen; BD Biosciences, Heidelberg, Germany) or anti-rabbit IgG (Santa Cruz Biotechnology, Inc., Dallas, TX, USA) conjugated to HRP, diluted 1:1000 in BSA 5% blocking buffer for 1 h at RT. Membranes were washed as above and revealed with the Clarity^TM^ western ECL detection reagent (Bio-Rad, Hercules, CA, USA) according to manufacturer’s instructions. Membranes were visualized on a ChemiDoc^TM^ touch gel imaging system (Bio-Rad, Hercules, CA, USA).

### 2.13. Statistical Analyses

Data analysis was conducted using the Stata version 17.0 (StataCorp LLC, College Station, TX, USA) and GraphPad Prism version 6 (2012 GraphPad Software Inc. La Jolla, CA, USA) software. Fisher’s exact test and Chi-square test were applied to compare two or three different categorical variables, respectively. For statistical analyses of cell experiments, Wilcoxon test, or Mann–Whitney U test were used. A *p*-value of 0.05 or less was considered statistically significant.

## 3. Results

### 3.1. Patients

Of the 102 FFPE samples, 9 were excluded for subsequent analysis because they resulted negative for β-globin fragment amplification. The median age of patients at diagnosis was 38 years (38.2 ± 9.7 years). Distribution according to histopathological classification was as follows: 22/93 (23.7%) LSIL, 52/93 (55.9%) HSIL, and 19/93 (20.4%) SCC. The occurrence of LSIL was more frequent in younger patients (≤38 years), while SCC was more common in older women (>38 years) (*p* = 0.001) (Table 1).

### 3.2. HPV Presence in Cervical Lesions

Generic HPV infection was evaluated by PCR using GP5+/GP6+ primers (Figure 1a, upper panel). HPV infection was detected in 6/22 (27.3%) LSIL, 38/52 (73.1%) HSIL, and 15/19 (78.9%) SCC cases. A statistically significant difference was obtained when the frequency of HPV infection in LSIL was compared to HSIL and SCCs (*p* < 0.001) (Table 2). Considering HPV positive samples, genotyping was successfully achieved by L1 fragment sequencing in 52/59 (88.1%) cases. In the remaining seven samples, HPV16 and 18 were evidenced in one sample each (1.7%) by specific PCR for each of these genotypes, while in five cases (8.5%), the HPV genotype was not identified (HPVX). The distribution of the most common HR-HPV genotypes was as follows: HPV16 (57.6%), HPV33 (8.5%), HPV31 (5.1%), HPV45 (5.1%), HPV18 (3.4%), and HPV58 (3.4%). Other HR-HPVs such as 35 and 66 were detected in one sample each (1.7%). HPV16 was only detected in HSIL and SCC, which was statistically significant compared to LSIL (*p* < 0.001). Low-risk HPVs 6 and 81 were evidenced in two (3.4%) and one (1.7%) positive tissues, respectively.

### 3.3. EBV Presence in Cervical Lesions

EBV presence was determined by PCR (Figure 1a, lower panel) and ISH (Figure 1b). By PCR, EBV DNA was detected in 16/22 (72.7%), 27/52 (51.9%), and 13/19 (68.4%) of LSIL, HSIL, and SCCs, respectively, with a non-statistically significant difference among groups (*p* = 0.177) (Table 3). A non-statistically significant difference between EBV positivity and patient age was also found (*p* = 0.529). To determine the lineage of EBV-infected cells, EBERs ISH was performed in all PCR EBV-positive tissue samples (Figure 1b). Overall, EBV was detected in 4/56 (7.1%) of cervical lesions by ISH. EBV infection in epithelial cells was confirmed in 2/56 (3.6%) of cases (one HSIL and one SCC). In one of these samples, an additional staining in tissue-infiltrating lymphocytes was evidenced (Figure 1b, lower right panel, arrow). In two other different cases (3.6%), EBV was only observed in tissue-infiltrating lymphocytes.

By PCR, HPV/EBV, and HR-HPV/EBV co-presence was detected in 36/93 (38.7%) and 30/93 (32.3%) cervical lesions, respectively. According to the type of lesion, HR-HPV/EBV co-presence was evidenced in 3/22 (13.6%) LSIL, 17/52 (32.7%) HSIL, and 11/19 (57.9%) SCCs with a statistically significant difference (*p* = 0.020) (Figure 1c). HPV16 subtype was evidenced in 18/30 (60.0%) of HR-HPV/EBV positive samples (Table 4, Figure 1d). EBV infection was also evidenced in 3/3 (100%) of LR-HPV infected samples. Absence of HPV/EBV co-presence was evidenced when EBV was assessed by ISH.

### 3.4. BARF1 Transcripts in EBV-Positive Cervical Pre-Malignant Lesions and Squamous Cell Carcinomas

The presence of BARF1 transcripts was evaluated in all EBV-positive samples tested by PCR. Overall, BARF1 transcripts were detected in 37/55 (67.3%) of these cases. According to the type of lesion, BARF1 transcripts were detected in 12/16 (75.0%) LSIL, 16/26 (61.5%) HSIL, and 9/13 (69.2%) SCCs, though without a statistically significant difference (*p* = 0.360, Table 5). The expression level of BARF1 was increased (fold change ≥ 0.41, median) in 18/36 (50.0%) positive samples. BARF1 was also detected in 19/30 (63.3%) HR-HPV/EBV co-infected tissues. cDNA for evaluating HPV16 E6 expression was available in sixteen HPV16/EBV positive cases. Co-expression of HPV16 E6 and BARF1 transcripts was evidenced in 6/16 (37.5%) of these cases.

### 3.5. BARF1 Increases Cervical Cancer Cell Proliferation

To determine the effects of BARF1 on cell proliferation rates, cervical cancer cell lines were stably transfected with a BARF1-encoding vector or an empty vector and BARF1 expression was corroborated in BARF1-transfected cells by RT-PCR (Figure 2a). BARF1- and empty vector-transfected cells were seeded, and proliferation was determined at 24, 48, 72, and 96 h using the MTS assay. As shown in Figure 2b (left), BARF1-expressing SiHa cells displayed an increased proliferation rate compared to empty vector, which was statistically significant at 48, 72, and 96 h (*p* = 0.002, *p* = 0.004, and *p* = 0.002, respectively). Similar results were obtained in CaSki cells, although the difference in proliferation rates between BARF1-transfected cells and empty vector was more evident than for SiHa cells (Figure 2b, right). The *p*-values for 48, 72, and 96 h were *p* = 0.002, *p* = 0.008. and *p* = 0.016, respectively.

### 3.6. BARF1 Promotes Cervical Cancer Cell Migration

The capacity of BARF1 to increase cervical cell migration was evaluated using the transwell migration assay. BARF1-expressing SiHa cells showed a statistically significant increase in migrated cells (180 ± 40), compared to empty vector-transfected cells (122 ± 19) (*p* = 0.009) (Figure 2c, left). Similar results were obtained for CaSki cells, with 122 ± 40 and 82 ± 5 for BARF1-expressing and empty vector-transfected cells, respectively (*p* = 0.002) (Figure 2c, right).

### 3.7. BARF1 Promotes Epithelial to Mesenchymal Transition in Cervical Cancer Cells

BARF1-expressing SiHa cells displayed morphological changes characterized by the loss of epithelial phenotype and the appearance of spindle-shaped cells (Figure 3a, right) compared to empty vector-transfected cells (Figure 3a, left). Thus, the levels of biomarkers related to epithelial to mesenchymal transition, such as E-cadherin and ZEB1, were assessed. As shown in Figure 3b, a statistically significant increase in the expression of ZEB1 protein was observed in BARF1-expressing SiHa cells compared to empty vector-transfected control cells (*p* = 0.050). Although a slight decrease in E-cadherin was observed in BARF1-transfected cells, the difference was not statistically significant. Similar results were obtained for CaSki cells (Figure 3c).

### 3.8. BARF1 Is Unable to Increase Anchorage-Independent Growth in Cervical Cancer Cells

The capacity of BARF1 to induce anchorage-independent growth was also assessed by counting the number of colony-forming cells in soft agar (Appendix A). The number of colonies was greater in BARF1-transfected SiHa cells (252.1 ± 120.2) compared to empty vector-transfected cells (221.2 ± 69.0), but no statistically significant difference was observed. Under the same experimental conditions, CaSki cells were unable to form colonies.

### 3.9. Phosphoproteomic NF-kB Assay in SiHa Cells

To evaluate which proteins could be potentially involved in the increased migration capacity of cells expressing BARF1, a phosphoproteomic assay for NF-κB signaling pathway activation was performed in SiHa cells. Some proteins such as p53, IRF5, STAT1 (pY701), c-Rel, and IL-18Ra were positively regulated in BARF1-expressing cells when compared to empty vector (Figure 4 and Appendix A). In contrast, NGFR/TNFRSF16, STAT2, ASC, STAT2 (pY689), and p53 (pS46), among others, were negatively regulated in the presence of BARF1.

## 4. Discussion

HR-HPV infection is a necessary condition for the development and progression of cervical carcinoma. However, only a small number of HR-HPV-infected women actually develop cervical cancer, suggesting that other cofactors are additionally required to promote this malignancy [15,16]. EBV infection has been detected in cervical carcinomas, suggesting that this persistent virus may be a cofactor for cervical cancer. Considering that both HR-HPV and EBV require additional cofactors for carcinogenesis [36,37], in the present study, we characterized HPV/EBV co-presence in low- and high-grade squamous intraepithelial lesions and squamous cervical carcinomas. Indeed, we evaluated the frequency of HPV/EBV co-presence in premalignant and malignant cervical lesions from Chilean women as well as its contribution to the progression of cervical cancer. HR-HPV infection rate was significantly increased in HSIL and cervical cancer when compared to LSIL, as previously reported [32,38]. Although it has been established that HR-HPV is present in almost 100% of cervical carcinomas, we were able to detect this infection in almost 80% of cases. Probably, factors related to sensitivity of the PCR methodology, DNA fragmentation in the clinical samples or a probably low viral load account for this percentage of detection. Indeed, Zehbe et al. reported that the sensitivity of this methodology (HPV detection using GP5+/GP6+ primers) is around 95% [39]. HPV16 was the predominant type encountered among HPV-positive specimens and its frequency increased along with the severity of cervical lesions. This result is consistent with data obtained by other authors [40,41], though interestingly, HPV33 was the second most frequent HPV genotype. HPV18, considered the second most frequent HR-HPV genotype worldwide, was only detected here in 3.4% of the cases.

In this study we determined by PCR that EBV is present in LSIL, HSIL and SCCs, as was previously reported using this methodology [42,43], although without a statistically significant difference. Though conventional PCR assay cannot distinguish between epithelial and lymphocyte infection, this result needs to be interpreted with caution because EBV was detected in one high grade and one cervical SCC by EBER1 ISH, considered the gold standard for EBV detection in clinical specimens. Sasagawa et al. detected less than 1 copy of EBV genome in cervical SCCs by PCR, which suggests a reduced number of cells infected with this virus. This fact was confirmed by BamHI-W mRNA in situ hybridization and IHC for LMP1 and EBNA2 proteins [42]. Unfortunately, the age of collected specimens may affect the sensitivity of ISH and could account for the low number of EBV ISH-positive cases detected in this study, as previously reported [44,45]. Nevertheless, HR-HPV/EBV co-presence by PCR was detected in 38.7% of cases, with a significant increase from LSIL to HSIL and cervical SCC. A similar pattern was previously reported by Khenchouche et al. [32]. In a meta-analysis conducted by de Lima et al., the pooled prevalence of HPV/EBV co-presence also increased from normal cervix, to LSIL, HSIL and SCC [26]. However, in aforementioned study the pooled prevalence for HR-HPV/EBV co-presence was not reported.

Interestingly, in the present work HPV16/EBV co-presence also showed a significant association with the grade of the cervical lesions. In previous studies, EBV was associated with infection by HR-HPV subtypes, such as HPV16, HPV18, or HPV31 [46], as well as with an increased risk of HPV16 integration into the host genome [47,48]. The co-presence of HPV and EBV was also reported in oropharyngeal squamous cell carcinomas [49], in which HPV16 is the most frequent genotype detected [50]. In addition, the presence of EBV DNA was evidenced in about 80% of HPV16 or HPV18 positive NPCs [51]. In fact, in head and neck cancer, EBV infection was significantly associated with HR-HPV subtypes such as HPV16, 18, 45, and 58 [52]. Moreover, the co-expression of HR-HPVs E6 and EBV LMP1 was related with advanced tumor stage [52]. Overall, these data suggest a potential role of EBV as a cofactor in the progression of cervical SCCs. Moreover, we cannot deny a potential role of EBV infection in tumor-infiltrating lymphocytes as an additional mechanism implicated in cervical cancer progression.

On the other hand, BARF1 was detected in most EBV-positive specimens (67.3%) and a semiquantitative analysis revealed the absence of a significant association with the cervical lesion grade. Although the expression of BARF1 is commonly found in NPC and EBVaGC, only one previous study reported experimental evidences of BARF1 protein expression in cervical cancer [32]. In fact, the expression of BARF1 protein was detected in frozen cervical SCCs using immunoblotting technique, which was also confirmed by immunohistochemistry. Interestingly, BARF1 protein expression was only detected in HPV+/EBV+ SCCs samples [32]. Additionally, in the present study the two EBV ISH-positive specimens showed BARF1 expression in tumor epithelial cells. In this regard, the role of BARF1 in development and progression of epithelial tumors was recently reviewed [53]. Moreover, HPV16 E6 and BARF1 transcripts were co-expressed in 37.5% of HPV16/EBV-positive cases, which suggests a potential cooperation of these proteins for cervical SCC progression, as was previously reported for HR-HPV E6 and EBV LMP1 protein co-expression [18].

In the present study, we also evaluated the capacity of BARF1 to induce phenotypic changes in HPV16-positive cervical cells. In fact, when SiHa and CaSki cells were transfected with a BARF1 encoding vector, the proliferation rate was significantly increased. In previous studies, it was demonstrated that BARF1 can increase the proliferation rate in a variety of epithelial cells [53]. For instance, treatment of human keratinocytes with exogenous BARF1 increased transition from G1 to S-phase of the cell cycle [54]. Transfection with BARF1 vector also increases the expression of cyclin D1 in the same cells [54]. Moreover, BARF1 induces proliferation of GC cells through a reduction in p21^WAF1^ [55] or NF-κB RelA upregulation [56]. Although in the present study we were unable to find NF-κB RelA activation in SiHa cells transfected with BARF1, the NF- κB subunit c-Rel was positively regulated in these cells, which is also related with the progression of a variety of epithelial tumors [57]. Furthermore, BARF1 was also able to increase the migration capacity of cervical cancer cells. Similarly, Hoebe et al. described alterations in the migration properties of BARF1-transfected epithelial cells [58].

In addition, we assessed the capacity of BARF1 to induce epithelial-mesenchymal transition (EMT) in cervical cancer cells. Notably, the expression of BARF1 was related to increased levels of ZEB1 protein, whereas E-cadherin expression decreased, albeit non-significantly. In a previous study, HPV16 E7 was shown to decrease the expression of E-cadherin through DNA methyltransferase 1 (DNMT1) overexpression [59]. Interestingly, McCormick et al. reported an increase in the methylation status of the E-cadherin gene (CDH1) in EBV-positive cervical lesions, although no statistical significance was found [27]. Overall, these data suggest a potential contribution of BARF1 to EMT in cervical cancer cells, although further studies investigating the DNMT1/E-Cadherin axis and/or the expression profile of other EMT-related molecules are strongly recommended. Moreover, BARF1 was unable to increase the anchorage-independent growth potential of SiHa cells. At present, the tumorigenic properties of BARF1 remain controversial. Previously, it was reported that BARF1 was able to induce anchorage-independent growth and tumor growth in nude mice in tumor and non-tumor epithelial cells [58,60]. In contrast, Jiang et al. found that BARF1 only increases the cell proliferation rate and the anchorage-independent growth of nasopharyngeal cancer cells in cooperation with H-ras [31]. Since SiHa cells contain 1 or 2 copies of HPV16, experiments in HPV-negative cervical cells transfected with BARF1 are necessary to elucidate the potential contribution of this molecule to tumorigenesis. In addition, the HPV-negative HaCaT cell line or primary human keratinocytes might represent alternative in vitro models for evaluating the role of BARF1 in epithelial carcinogenesis, as was previously reported [54,61].

In summary, we report here that EBV is present in preneoplastic lesions as well as in cervical carcinomas, without a significant difference. Additionally, BARF1 is expressed in a majority of EBV positive cases, which in turn can increase proliferation, migration and EMT of cervical cancer cells in vitro. More studies are warranted to dissect the role of EBV and BARF1 in cervical cancer.

## Figures and Tables

**Figure 1 microorganisms-10-00888-f001:**
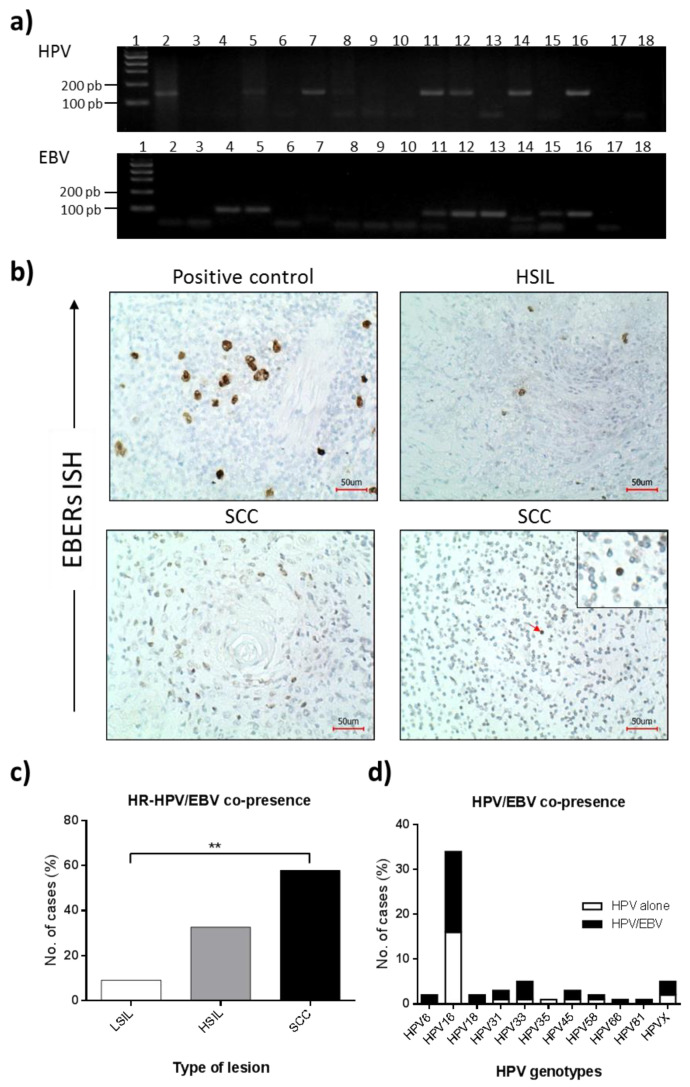
HPV and EBV infection in cervical lesions. (**a**) Conventional PCR for HPV (upper panel) and EBV (lower panel). Lane 1: 100-pb DNA ladder, Lanes 2–15: cervical samples, Lane 16: positive control (HeLa cells), Lanes 17 and 18: negative controls (extraction buffer or nuclease-free water). (**b**) In situ hybridization for EBERs in cervical lesions. A Hodgkin’s lymphoma sample was used as positive control. A positive signal for EBERs (brown color) is observed in a HSIL and a cervical SCC sample, located in epithelial cells and tumor-infiltrating lymphocytes (red arrow). Samples were counterstained with Hematoxylin (blue color). Red bar = 50 µm. (**c**) HR-HPV/EBV co-presence was increased in cervical SCC when compared to LSIL (** *p* = 0.008). (**d**) Distribution of HPV/EBV co-presence according to HPV genotype.

**Figure 2 microorganisms-10-00888-f002:**
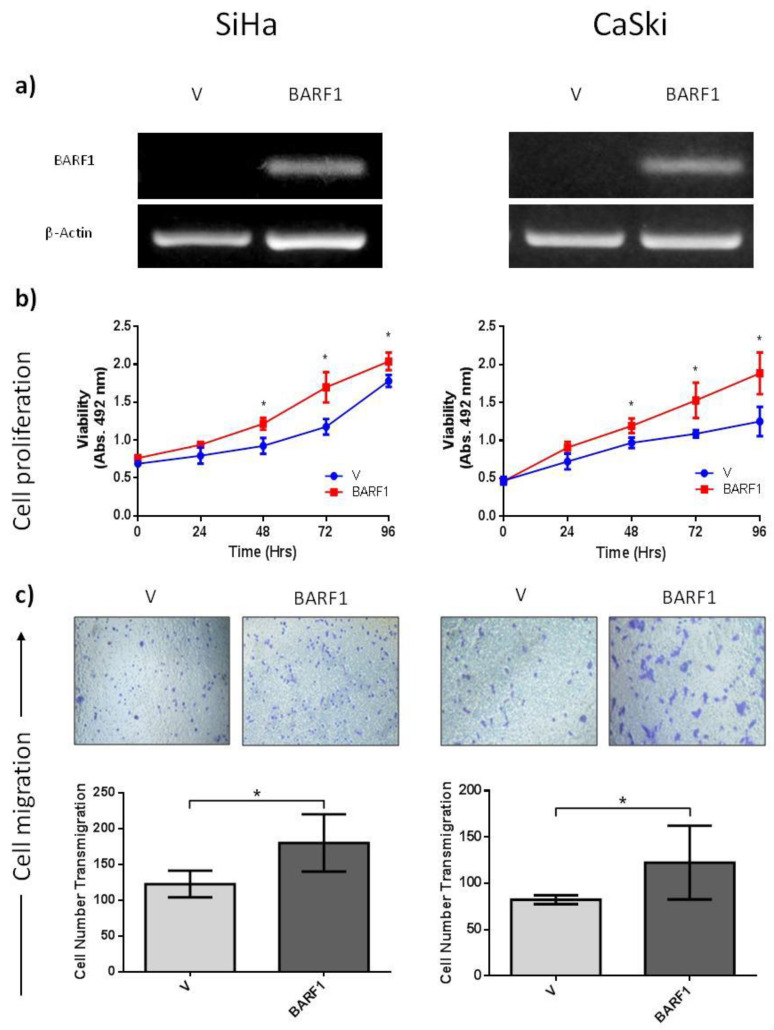
BARF1 increases cell proliferation and migration of HPV16-positive cervical cancer cells. (**a**) RT-PCR to evaluate BARF1 transcript levels in SiHa and CaSki cells transfected with empty vector (V) or BARF1 vector; β-actin was used as loading control. (**b**) Cell proliferation assay performed in SiHa and CaSki cells. For BARF1-transfected SiHa cells, statistically significant differences were obtained at 48, 72, and 96 h of culture (* *p* = 0.002, * *p* = 0.004, and * *p* = 0.002, respectively) compared to empty vector-transfected cells. Statistically significant differences were also evidenced for BARF1-expressing CaSki cells at 48, 72, and 96 h of culture (* *p* = 0.002, * *p* = 0.008, and * *p* = 0.016, respectively). (**c**) Transwell migration assay performed in SiHa and CaSki cells transfected with BARF1 or empty vector. Increased migration was evidenced in both SiHa and CaSki cells transfected with BARF1 after 7 h of culture using fibronectin-coated transwell inserts, compared to empty vector-transfected cells (* *p* = 0.009 and * *p* = 0.002, respectively).

**Figure 3 microorganisms-10-00888-f003:**
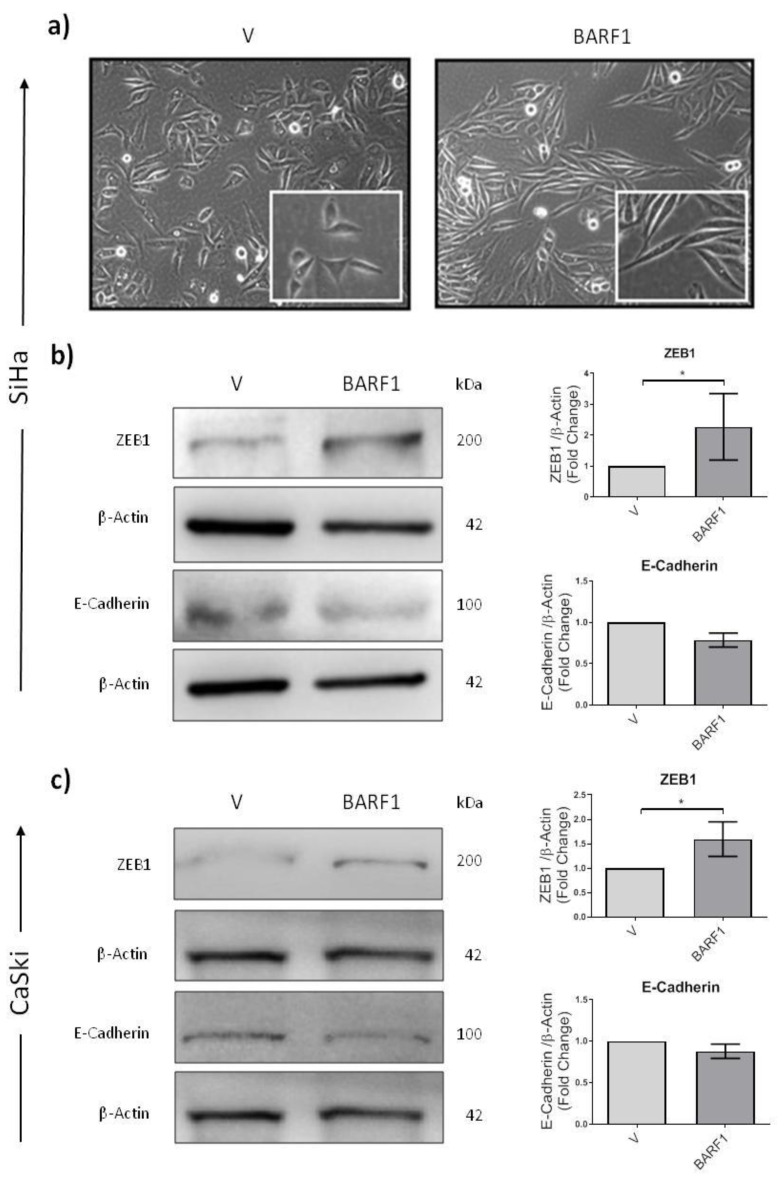
BARF1 promotes epithelial to mesenchymal transition in cervical cells. (**a**) Morphological changes characterized by the appearance of spindle-shaped cells were observed in SiHa cells stably transfected with BARF1 compared to empty vector-transfected cells. Analysis of ZEB1 and E-cadherin protein levels in SiHa and CaSki cells transfected with BARF1 or empty vector (**b**,**c**, **left**). A significant increase in the expression of ZEB1 was evidenced in BARF1-expressing SiHa and CaSki cells compared to empty vector-transfected cells (**b**,**c**, **right**) (* *p* = 0.050, for both). A decrease in E-cadherin was also evidenced in BARF1 transfected cells, but not statistically significant. Densitometric analyses of three independent assays normalized against β-actin were plotted. Data are presented as mean ± SEM.

**Figure 4 microorganisms-10-00888-f004:**
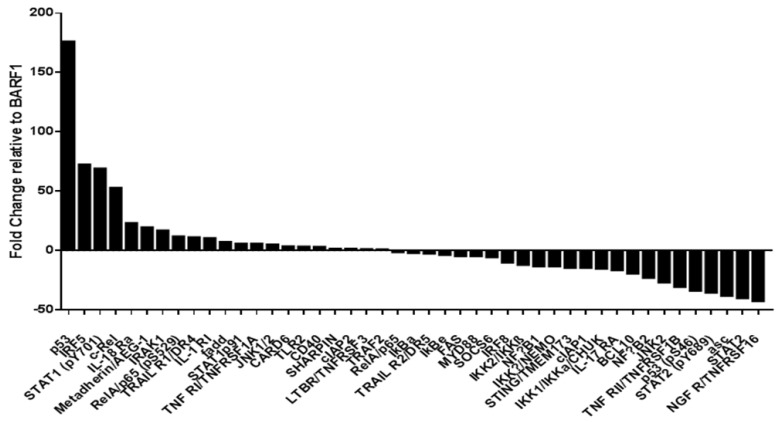
Protein array for NF-κB signaling pathway activation in SiHa cells. Activation of NF-κB signaling pathway was compared between SiHa cells transfected with BARF1 or empty vector. Data were presented in reference to the fold-change occurring in the presence of BARF1 compared to the empty vector control.

**Table 1 microorganisms-10-00888-t001:** Clinicopathological features of patients.

Type of Lesion		Age at Diagnosis		
	Total	≤38	>38	*p*-Value
	No. Cases	No. (%)	No. (%)	
LSIL	22	16 (72.7)	6 (27.3)	0.001 ^a^
HSIL	52	28 (53.8)	24 (46.2)	
SCC	19	3 (18.8)	16 (84.2)	

^a^ The increase in SCC frecuency in patients over 38 years old and in LSIL frequency in patients younger than 38 was statistically significant.

**Table 2 microorganisms-10-00888-t002:** Frequency of generic HPV and HPV16 according to the type of cervical lesion.

Type of Lesion		Generic HPV				HPV16		
	Total	HPV−	HPV+	*p*-Value	Total	HPV16−	HPV16+	*p*-Value
	No. Cases	No. (%)	No. (%)		No. Cases	No. (%)	No. (%)	
LSIL	22	16 (72.7)	6 (27.3)	<0.001 ^a^	22	22 (100)	0 (0)	<0.001 ^b^
HSIL	52	14 (26.9)	38 (73.1)		52	29 (55.8)	23 (44.2)	
SCC	19	4 (21.1)	15 (78.9)		19	8 (42.1)	11 (57.9)	

^a,b^ The increased frequency of generic HPV or HPV16 infection in HSIL and SCC compared to LSIL is statistically significant.

**Table 3 microorganisms-10-00888-t003:** Frequency of EBV infection according to grade of cervical lesions.

Type of Lesion		EBV/PCR			EBV/ISH(Epithelial Cells)		
	Total	EBV−	EBV+	*p*-Value	EBV−	EBV+	*p*-Value
	No. Cases	No. (%)	No. (%)		No. (%)	No. (%)	
LSIL	22	6 (27.3)	16 (72.7)	0.177 ^a^	16 (100)	0 (0)	0.539 ^b^
HSIL	52	25 (48.1)	27 (51.9)		26 (98.1)	1 (1.9)	
SCC	19	6 (31.6)	13 (68.4)		12 (94.7)	1 (5.3)	

^a,b^ The presence of EBV measured by means of PCR or ISH was not related with the grade of cervical lesions.

**Table 4 microorganisms-10-00888-t004:** Frequency of HPV16/EBV coinfection in cervical lesions.

HPV16 Detection		EBV Detected by PCR		
	Total	EBV−	EBV+	*p*-Value
	No. Cases	No. (%)	No. (%)	
HPV16+	34	16 (47.1)	18 (52.9)	0.277 ^a^
HPV16−	59	21 (35.6)	38 (64.4)	

^a^ No statistically significant difference was obtained between EBV detected by PCR and HPV16 infection.

**Table 5 microorganisms-10-00888-t005:** Expression of BARF1 according to type of lesion.

Type of Lesion		BARF1 Positivity				BARF1 Expression Level		
	Total	BARF1−	BARF1+	*p*-Value	Total	BARF1 Low	BARF1 High	*p*-Value
	No. Cases	No. (%)	No. (%)		No. Cases	No. (%)	No. (%)	
LSIL	16	4 (25.0)	12 (75.00)	0.655 ^a^	12	4 (33.3)	8 (66.7)	0.360 ^b^
HSIL	26	10 (38.5)	16 (61.5)		15 *	9 (60.0)	6 (40.0)	
SCC	13	4 (30.8)	9 (69.2)		9	5 (55.6)	4 (44.4)	

^a,b^ No statistically significant differences were obtained when BARF1 positivity or BARF1 expression levels were compared with the type of cervical lesion. * β-actin control was not available for one BARF1-positive sample.

## Data Availability

Not applicable.

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
