# Peer review of "Characterization of High-Risk HPV/EBV Co-Presence in Pre-Malignant Cervical Lesions and Squamous Cell Carcinomas"

_microorganisms, 2022, doi:10.3390/microorganisms10050888_

Round 1
Reviewer 1 Report
The study entitled “Characterization of High-risk HPV/EBV Co-presence in Premalignant Cervical Lesions and Squamous cell carcinomas” (Manuscript ID: microorganisms-1575909) by Dr. Blanco evaluated the prevalence of high-risk HPV and EBV co-presence in pre-tumoral lesions cervical carcinomas from Chilean patients., while the expression of EBV BARF1 was also assessed. In my opinion, this is an interesting work and well organized and with a well performed experimental design. Despite too verbose in several sections (methods) the ms reports interesting data on the role of HPV/EBV coinfection upon cervical carcinogenesis.
I therefore recommend a major revision. I have several comments for improving the manuscript
General comments
- In the methods the two cohorts of patients should be detailed, e.g., number, median age, presence of comorbidities etcc. In addition, inclusion and exclusion criteria should be detailed
- The methods are too verbose. I suggest reorganizing the text by reducing it for at least 30% by moving several sections of the methods to the supplemental material. For instance, primers sequences can be summarized in a supplemental table.
- Methos are entirely lacking in supporting references. Please include references.
- Tables it is difficult to understand for which comparison the p values are referred to
- The statistical methods employed should be moved from the result section
- Several panels from figure 3 should be enlarged as difficult to read
- The authors should describe in the discussion the main the differences between the results obtained in the present study and those reported previously by Lima, M.A.P., et al., Gynecol Oncol, 2018. 549 148(2): p. 317-328.
Minor
Line 25 better “(HR-HPVs)”
Lines 27-28 please check the word colors
Line 30 The number of females should be detailed
Line 35 the rational behind the selection of BARP1 should be included in the abstract (i.e., it is a EBV lytic gene)
Line 55 For com’pletness I suggest including additional lethal HPV-driven carcinomas of the genital tract, such as vulvar (DOI: 10.1097/PAP.0000000000000155 ) and penile (doi: 10.3892/etm.2019.8181) carcinomas
Line 59 HPV16/HPV18 types also play an important role in pre-neoplastic lesion (CIN) tumorigenesis (DOI: 10.3389/fmicb.2020.591452). this information/reference should be included
Line 139 better “viral DNA sequencing”
Line 285 difference among who?
Lines 487-489 HPV-negative HaCaT cell line might represent a helpful in vitro model for evaluating the orle of BARF1 in cervical carcinogenesis
Reviewer 2 Report
This study examines whether HPV positive cervical cell cancers or premalignant lesions are co-infected with Epstein-Barr virus (EBV) using three different methods (DNA PCR and EBER in situ hybridization, and RT-PCR for EBV BARF1 gene). Depending upon the methods used, they get different results. The "gold standard" method for detecting EBV, in situ hybridization to detect the highly expressed EBV EBERs RNA, only finds EBV infected cells in a few HPV+ squamous cell carcinomas and high grade lesions, and many of the rare EBV+ cells appear to be B cells trafficking through the lesions. Nevertheless, the authors do show what appear to be a fair number of authentic EBV+ epithelial cells in at least one HPV+ cervical cancer which would at least document that some HPV+ cervical cancers are also con-infected with EBV. When the authors perform PCR studies or do RNA-PCR to look for expression of the EBV BARF1 transcript (an EBV gene which may be highly expressed in epithelial cells), they find that a much higher percentage (up to 60%) of the tumors or premalignant lesions have EBV DNA and/or EBV transcript. The problem with the latter two approaches is the over-sensitivity of the methods (such that infection of even a few B cells in the tumors could give a positive result) and the potential for contamination. In addition, it is not clear to this reviewer what the appropriate negative control samples are being used by these investigators. The last part of the paper concentrates on studying the phenotype of BARF1 over-expression in HPV positive cervical cancer cell lines in vitro; however, these studies seem a little premature if BARF1 is not actually expressed in HPV-infected cervical carcinomas.
A positive aspect of this paper is that it is well written and the authors are honest about the limitations of their study. The paper would be improved by either of the following: 1) showing that the BARF1 protein is expressed in cervical carcinomas that supposedly express the BARF1 transcript; and/or 2) looking at the RNA-seq data sets available in the public domain for HPV+ cervical cancers to determine if the BARF1 transcript is detected in these tumors. This reviewer's understanding is that such public RNA-seq data sets have not shown evidence of EBV transcripts at levels higher than expected due to rare EBV+ infected B cells trafficking through the tumors, but perhaps specific examination of BARF1 transcripts would give a different answer.
Reviewer 3 Report
Blanco et al. describe a sample of cervical lesions, both low and high grade, and investigate the presence of both HPV and EBV, aiming to deduce whether coinfection is associated with progression of cervical cancer. They go on to investigate EBV transcript BARF in these samples, and aim to dissect a role in invasion/EMT phenotypes of aggressive cancers. The article is sound, I have just a few points to be addressed in the discussion portion of the paper:
- Overall, this work agrees with established research and, although the authors’ focus is on cervical lesions, there are several studies focusing on EBV and HPV in oropharyngeal SCC, which it may be interesting to include in the discussion. Especially as these cancers are known to be predominantly HPV16-driven.
- The HPV16-positive lines, SiHa and Caski, are cancer-derived cell lines. As they are already transformed, it may be interesting for the authors to look at the effect of BARF (or indeed EBV as a whole virus) in models of earlier stages of the HPV tumorigenesis, for example in primary keratinocytes.
- Although there may be a trend to decrease E-Cadherin, it is not statistically significant so I don’t think any conclusions can be drawn. It may be worth investigating DNMT1/E-Cadherin axis a little further in the EBV/HPV context to further dissect this.
Author Response
Pending revision.

Round 2
Reviewer 1 Report
The authors have responded appropriately to the reviewers' comments.